# TENSOR DECOMPOSITIONS FOR TEMPORAL KNOWLEDGE BASE COMPLETION

**Timothee Lacroix[1,2], Guillaume Obozinski[3], Nicolas Usunier[1]**
[1] Facebook AI Research  [2] ENPC[*]  [3] Swiss Data Science Center, EPFL & ETH Zürich
`timothee.lax@gmail.com`  `guillaume.obozinski@epfl.ch`
`usunier@fb.com`

## ABSTRACT

Most algorithms for representation learning and link prediction in relational data have been designed for static data. However, the data they are applied to usually evolves with time, such as friend graphs in social networks or user interactions with items in recommender systems. This is also the case for knowledge bases, which contain facts such as (US, has president, B. Obama, [2009-2017]) that are valid only at certain points in time. For the problem of link prediction under temporal constraints, i.e., answering queries such as (US, has president, ?, 2012), we propose a solution inspired by the canonical decomposition of tensors of order 4. We introduce new regularization schemes and present an extension of ComplEx (Trouillon et al., 2016) that achieves state-of-the-art performance. Additionally, we propose a new dataset for knowledge base completion constructed from Wikidata, larger than previous benchmarks by an order of magnitude, as a new reference for evaluating temporal and non-temporal link prediction methods.

## 1  INTRODUCTION

Link prediction in relational data has been the subject of interest, given the widespread availability of such data and the breadth of its use in bioinformatics (Zitnik et al., 2018), recommender systems (Koren et al., 2009) or Knowledge Base completion (Nickel et al., 2016a). Relational data is often temporal, for example, the action of buying an item or watching a movie is associated to a timestamp. Some medicines might not have the same adverse side effects depending on the subject's age. The task of *temporal* link prediction is to find missing links in graphs at precise points in time.

In this work, we study temporal link prediction through the lens of temporal knowledge base completion, which provides varied benchmarks both in terms of the underlying data they represent, but also in terms of scale. A knowledge base is a set of facts (subject, predicate, object) about the world that are known to be true. Link prediction in a knowledge base amounts to answer incomplete queries of the form (subject, predicate, ?) by providing an accurate ranking of potential objects. In temporal knowledge bases, these facts have some temporal metadata attached. For example, facts might only hold for a certain time interval, in which case they will be annotated as such. Other facts might be event that happened at a certain point in time. Temporal link prediction amounts to answering queries of the form (subject, predicate, ?, timestamp). For example, we expect the ranking of queries (USA, president, ?, timestamp) to vary with the timestamps.

As tensor factorization methods have proved successful for Knowledge Base Completion (Nickel et al., 2016a; Trouillon et al., 2016; Lacroix et al., 2018), we express our Temporal Knowledge Base Completion problem as an order 4 tensor completion problem. That is, timestamps are discretized and used to index a 4-th mode in the binary tensor holding (subject, predicate, object, timestamps) facts.

First, we introduce a ComplEx (Trouillon et al., 2016) decomposition of this order 4 tensor, and link it with previous work on temporal Knowledge Base completion. This decomposition yields embeddings for each timestamps. A natural prior is for these timestamps representation to evolve slowly over time. We are able to introduce this prior as a regularizer for which the optimum is a

---

[*]Université Paris-Est, Equipe Imagine, LIGM (UMR8049) Ecole des Ponts ParisTech, Marne-la-Vallée

variation on the nuclear $p$-norm. In order to deal with heterogeneous temporal knowledge bases where a significant amount of relations might be non-temporal, we add a non-temporal component to our decomposition.

Experiments on available benchmarks show that our method outperforms the state of the art for similar number of parameters. We run additional experiments for larger, regularized models and obtain improvements of up to 0.07 absolute Mean Reciprocal Rank (MRR).

Finally, we propose a dataset of $400k$ entities, based on Wikidata, with $7M$ train triples, of which $10\%$ contain temporal validity information. This dataset is larger than usual benchmarks in the Knowledge Base completion community and could help bridge the gap between the method designed and the envisaged web-scale applications.

## 2    RELATED WORK

Matrices and tensors are upper case letters. The $i$-th row of $U$ is denoted by $u_i$ while it's $j-th$ column is denoted by $U_{:,j}$. The tensor product of two vectors is written $\otimes$ and the hadamard (elementwise) product $\odot$.

**Static link prediction methods**    Standard tensor decomposition methods have lead to good results (Yang et al., 2014; Trouillon et al., 2016; Lacroix et al., 2018; Balažević et al., 2019) in Knowledge Base completion. The Canonical Polyadic (CP) Decomposition (Hitchcock, 1927) is the tensor equivalent to the low-rank decomposition of a matrix. A tensor $X$ of canonical rank $R$ can be written as:

$$X = \sum_{r=1}^{R} U_{:,r} \otimes V_{:,r} \otimes W_{:,r} = [\![U, V, W]\!] \iff \forall (i, j, k),\ X_{i,j,k} = \sum_{r=1}^{R} u_{i,r} v_{j,r} w_{k,r} = \langle u_i, v_j, w_k \rangle$$

Setting $U = W$ leads to the Distmult (Yang et al., 2014) model, which has been successful, despite only being able to represent symmetric score functions. In order to keep the parameter sharing scheme but go beyond symmetric relations, Trouillon et al. (2016) use complex parameters and set $W$ to the complex conjugate of $U$, $\overline{U}$. Regularizing this algorithm with the variational form of the tensor nuclear norm as well as a slight transformation to the learning objective (also proposed in Kazemi & Poole (2018)) leads to state of the art results in Lacroix et al. (2018).

Other methods are not directly inspired from classical tensor decompositions. For example, TransE (Bordes et al., 2013) models the score as a distance of the translated subject to an object representation. This method has lead to many variations (Ji et al., 2015; Nguyen et al., 2016; Wang et al., 2014), but is limited in the relation systems it can model (Kazemi & Poole, 2018) and does not lead to state of the art performances on current benchmarks. Finally Schlichtkrull et al. (2018) propose to generate the entity embeddings of a CP-like tensor decomposition by running a forward pass of a Graph Neural Network over the training Knowledge Base. The experiments included in this work did not lead to better link prediction performances than the same decomposition (Distmult) directly optimized (Kadlec et al., 2017).

**Temporal link prediction methods**    Sarkar & Moore (2006) describes a bayesian model and learning method for representing temporal relations. The temporal smoothness prior used in this work is similar to the gradient penalty we describe in Section 3.3. However, learning one embedding matrix per timestamp is not applicable to the scales considered in this work. Bader et al. (2007) uses a tensor decomposition called ASALSAN to express temporal relations. This decomposition is related to RESCAL (Nickel et al., 2011) which underperforms on recent benchmarks due to overfitting (Nickel et al., 2016b).

For temporal knowledge base completion, Goel et al. (2020) learns entity embeddings that change over time, by masking a fraction of the embedding weights with an activation function of learned frequencies. Based on the Tucker decomposition, ConT (Ma et al., 2018) learns one new core tensor for each timestamp. Finally, viewing the time dimension as a sequence to be predicted, García-Durán et al. (2018) use recurrent neural nets to transform the embeddings of standard models such as TransE or Distmult to accomodate the temporal data.

| DE-SimplE | $2r\left((3\gamma + (1-\gamma))|E| + |P|\right)$ |
|-----------|------------------------------------------------|
| TComplEx  | $2r(|E| + |T| + 2|P|)$                         |
| TNTComplEx | $2r(|E| + |T| + 4|P|)$                        |

Table 1: Number of parameters for each models considered

This work follows Lacroix et al. (2018) by studying and extending a regularized CP decomposition of the training set seen as an order 4 tensor. We propose and study several regularizer suited to our decompositions.

## 3    MODEL

In this section, we are given facts (subject, predicate, object) annotated with timestamps, we discretize the timestamp range (eg. by reducing timestamps to years) in order to obtain a training set of 4-tuple (subject, predicate, object, time) indexing an order 4 tensor. We will show in Section 5.1 how we reduce each datasets to this setting. Following Lacroix et al. (2018), we minimize, for each of the train tuples $(i, j, k, l)$, the instantaneous multiclass loss :

$$\ell(\hat{X}; (i,j,k,l)) = -\hat{X}_{i,j,k,l} + \log\left(\sum_{k'} \exp\left(\hat{X}_{i,j,k',l}\right)\right).\tag{1}$$

Note that this loss is only suited to queries of the type (subject, predicate, ?, time), which is the queries that were considered in related work. We consider another auxiliary loss in Section 6 which we will use on our Wikidata dataset. For a training set $S$ (augmented with reciprocal relations (Lacroix et al., 2018; Kazemi & Poole, 2018)), and parametric tensor estimate $\hat{X}(\theta)$, we minimize the following objective, with a *weighted* regularizer $\Omega$:

$$\mathcal{L}(\hat{X}(\theta)) = \frac{1}{|S|}\sum_{(i,j,k,l)\in S}\left[\ell(\hat{X}(\theta); (i,j,k,l)) + \lambda\Omega(\theta; (i,j,k,l))\right].$$

The ComplEx (Trouillon et al., 2016) decomposition can naturally be extended to this setting by adding a new factor $T$, we then have:

$$\hat{X}(U,V,T) = \mathrm{Re}\left(\llbracket U,V,\overline{U},T\rrbracket\right) \iff \hat{X}(U,V,T)_{i,j,k,l} = \mathrm{Re}\left(\langle u_i, v_j, \overline{u_k}, t_l\rangle\right)\tag{2}$$

We call this decomposition TComplEx. Intuitively, we added timestamps embedding that modulate the multi-linear dot product. Notice that the timestamp can be used to equivalently modulate the objects, predicates or subjects to obtain time-dependent representation:

$$\langle u_i, v_j, \overline{u_k}, t_l\rangle = \langle u_i \odot t_l, v_j, \overline{u_k}\rangle = \langle u_i, v_j \odot t_l, \overline{u_k}\rangle = \langle u_i, v_j, \overline{u_k} \odot t_l\rangle.$$

Contrary to DE-SimplE (Goel et al., 2020), we do not learn temporal embeddings that scale with the number of entities (as frequencies and biases), but rather embeddings that scale with the number of timestamps. The number of parameters for the two models are compared in Table 1.

### 3.1    NON-TEMPORAL PREDICATES

Some predicates might not be affected by timestamps. For example, Malia and Sasha will always be the daughters of Barack and Michelle Obama, whereas the "has occupation" predicate between two entities might very well change over time. In heterogeneous knowledge bases, where some predicates might be temporal and some might not be, we propose to decompose the tensor $\hat{X}$ as the sum of two tensors, one temporal, and the other non-temporal:

$$\hat{X} = \mathrm{Re}\left(\llbracket U,V^t,\overline{U},T\rrbracket + \llbracket U,V,\overline{U},\mathbf{1}\rrbracket\right) \iff \hat{X}_{i,j,k,l} = Re\left(\langle u_i, v_j^t \odot t_l + v_j, \overline{u_k}\rangle\right)\tag{3}$$

We call this decomposition TNTComplEx. Goel et al. (2020) suggests another way of introducing a non-temporal component, by only allowing a fraction $\gamma$ of components of the embeddings to be modulated in time. By allowing this sharing of parameters between the temporal and non-temporal part of the tensor, our model removes one hyperparameter. Moreover, preliminary experiments showed that this model outperforms one without parameter sharing.

## 3.2 REGULARIZATION

Any order 4 tensor can be considered as an order 3 tensor by *unfolding* modes together. For a tensor $X \in \mathbb{R}^{N_1 \times N_2 \times N_3 \times N_4}$, unfolding modes 3 and 4 together will lead to tensor $\tilde{X} \in \mathbb{R}^{N_1 \times N_2 \times N_3 N_4}$ (Kolda & Bader, 2009).

We can see both decompositions ((2) and (3)) as order 3 tensors by unfolding the temporal and predicate modes together. Considering the decomposition implied by these unfoldings (see Appendix 8.1) leads us to the following weighted regularizers (Lacroix et al., 2018):

$$\Omega^3(U, V, T; (i, j, k, l)) = \frac{1}{3} \left( \|u_i\|_3^3 + \|u_k\|_3^3 + \|v_k \odot t_l\|_3^3 \right) \tag{4}$$

$$\Omega^3(U, V^t, V, T; (i, j, k, l)) = \frac{1}{3} \left( 2\|u_i\|_3^3 + 2\|u_k\|_3^3 + \|v_j^t \odot t_l\|_3^3 + \|v_j\|_3^3 \right)$$

The first regularizer weights objects, predicates and pairs (predicate, timestamp) according to their respective marginal probabilities. This regularizer is a variational form of the weighted nuclear 3-norm on an order 4 tensor (see subsection 3.4 and Appendix 8.3 for details and proof). The second regularizer is the sum of the nuclear 3 penalties on tensors $[\![U, V^t, \overline{U}, T]\!]$ and $[\![U, V, \overline{U}]\!]$.

## 3.3 SMOOTHNESS OF TEMPORAL EMBEDDINGS

We have more a priori structure on the temporal mode than on others. Notably, we expect smoothness of the application $i \mapsto t_i$. In words, we expect neighboring timestamps to have close representations. Thus, we penalize the norm of the discrete derivative of the temporal embeddings :

$$\Lambda_p(T) = \frac{1}{|T| - 1} \sum_{i=1}^{|T|-1} \|t_{i+1} - t_i\|_p^p. \tag{5}$$

We show in Appendix 8.2 that the sum of $\Lambda_p$ and the variational form of the nuclear $p$ norm (6) lead to a variational form of a new tensor atomic norm.

## 3.4 NUCLEAR $p$-NORMS OF TENSORS AND THEIR VARIATIONAL FORMS

As was done in Lacroix et al. (2018), we aim to use tensor nuclear $p$-norms as regularizers. The definition of the nuclear $p$-norm of a tensor (Friedland & Lim, 2018) of order $D$ is:

$$\|X\|_{p*} = \inf_{\alpha, R, U^{(1)}, \dots, U^{(D)}} \left\{ \|\alpha\|_1 \mid X = \sum_{r=1}^{R} \alpha_r U_{:,r}^{(1)} \otimes \cdots \otimes U_{:,r}^{(D)}, \forall r, d \; \|U_{:,r}^{(d)}\|_p = 1 \right\}.$$

This formulation of the nuclear $p$-norm writes a tensor as a sum over *atoms* which are the rank-1 tensors of unit $p$-norm factors. The nuclear $p$-norm is NP-hard to compute (Friedland & Lim, 2018). Following Lacroix et al. (2018), a practical solution is to use the equivalent formulation of nuclear $p$-norm using their *variational form*, which can be conveniently written for $p = D$:

$$\|X\|_{D*} = \frac{1}{D} \inf_{X=[\![U^{(1)}, \dots, U^{(D)}]\!]} \sum_{d=1}^{D} \sum_{r=1}^{R} \|U_{:,r}^{(d)}\|_D^D. \tag{6}$$

For the equality above to hold, the infimum should be over all possible $R$. The practical solution is to fix $R$ to the desired rank of the decomposition. Using this variational formulation as a regularizer leads to state of the art results for order-3 tensors (Lacroix et al., 2018) and is convenient in a stochastic gradient setting because it separates over each model coefficient.

In addition, this formulation makes it easy to introduce a weighting as recommended in Srebro & Salakhutdinov (2010); Foygel et al. (2011). In order to learn under non-uniform sampling distributions, one should penalize the weighted norm : $\| \left( \sqrt{M^{(1)}} \otimes \sqrt{M^{(2)}} \right) \odot X \|_{2*}$, where $M^{(1)}$ and $M^{(2)}$ are the empirical row and column marginal of the distribution. The variational form (6) makes this easy, by simply penalizing rows $U_{i_1}^{(1)}, \ldots, U_{i_D}^{(D)}$ for observed triple $(i_1, \ldots, i_D)$ in stochastic gradient descent. More precisely for $D = 2$ and $N^{(d)}$ the vectors holding the observed count of each index over each mode $d$:

$$\frac{1}{|S|} \sum_{(i,j) \in S} \|u_i\|_2^2 + \|v_j\|_2^2 = \sum_i \frac{N_i^{(1)}}{S} \|u_i\|_2^2 + \sum_j \frac{N_j^{(2)}}{S} \|v_j\|_2^2 = \sum_i M_i^{(1)} \|u_i\|_2^2 + \sum_j M_j^{(2)} \|v_j\|_2^2.$$

In subsection 3.3, we add another penalty in Equation (5) which changes the norm of our atoms.In subsection 3.2, we introduced another variational form in Equation (4) which allows to easily penalize the nuclear 3-norm of an order 4 tensor. This regularizer leads to different weighting. By considering the unfolding of the timestamp and predicate modes, we are able to weight according to the joint marginal of timestamps and predicates, rather than by the product of the marginals. This can be an important distinction if the two are not independent.

## 3.5 Experimental impact of the regularizers

We study the impact of regularization on the ICEWS05-15 dataset, for the TNTComplEx model. For details on the experimental set-up, see Section 5.1. The first effect we want to quantify is the effect of the regularizer $\Lambda_p$. We run a grid search for the strength of both $\Lambda_p$ and $\Omega^3$ and plot the convex hull as a function of the temporal regularization strength. As shown in Figure 1, imposing smoothness along the time mode brings an improvement of over 2 MRR point.

The second effect we wish to quantify is the effect of the choice of regularizer $\Omega$. A natural regularizer for TNTComplEx would be :

$$\Delta^p(U, V, T; (i, j, k, l)) = \frac{1}{p} \left( 2\|u_i\|_p^p + 2\|u_k\|_p^p + \|v_j^t\|_p^p + \|t_l\|_p^p + \|v_j\|_p^p \right).$$

We compare $\Delta^4$, $\Delta^3$ and $\Delta^2$ with $\Omega^3$. The comparison is done with a temporal regularizer of $0$ to reduce the experimental space.

$\Delta^2$ is the common weight-decay frequently used in deep-learning. Such regularizers have been used in knowledge base completion (Nickel et al., 2011; 2016b; Trouillon et al., 2016), however, Lacroix et al. (2018) showed that the infimum of this penalty is non-convex over tensors.

$\Delta^3$ matches the order used in the $\Omega^3$ regularizer, and in previous work on knowledge base completion (Lacroix et al., 2018). However, by the same arguments, its minimization does not lead to a convex penalty over tensors.

$\Delta^4$ is the sum of the variational forms of the Nuclear 4-norm for the two tensors of order 4 in the TNTComplEx model according to equation (6).

Detailed results of the impact of regularization on the performances of the model are given in Figure 1. The two regularizers $\Delta^4$ and $\Omega^3$ are the only regularizers that can be interpreted as sums of tensor norm variational forms and perform better than their lower order counterparts.

There are two differences between $\Delta^4$ and $\Omega^3$. First, whereas the first is a variational form of the nuclear 4-norm, the second is a variational form of the nuclear 3-norm which is closer to the nuclear 2-norm. Results for exact recovery of tensors have been generalized to the nuclear 2-norm, and to the extent of our knowledge, there has been no formal study of generalization properties or exact recovery under the nuclear $p$-norm for $p$ greater than two.

Second, the weighting in $\Delta^4$ is done separately over timestamps and predicates, whereas it is done jointly for $\Omega^3$. This leads to using the joint empirical marginal as a weighting over timestamps and predicates. The impact of weighting on the guarantees that can be obtained are described more precisely in Foygel et al. (2011).

The contribution of all these regularizers over a non-regularized model are summarized in Table 3. Note that careful regularization leads to a 0.05 MRR increase.

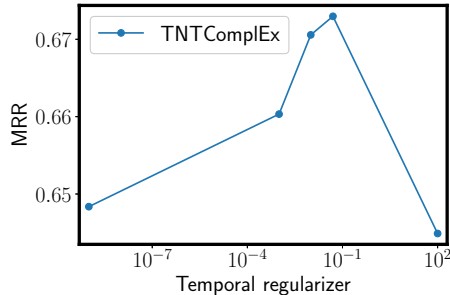 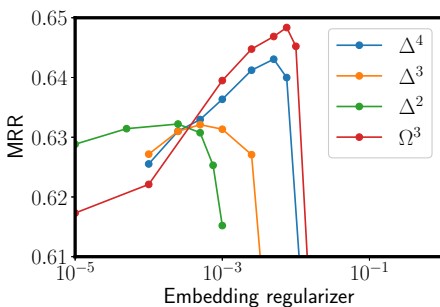

Figure 1: Impact of the temporal (left) regularizer and embeddings (right) regularizer on a TNT-ComplEx model trained on ICEWS05-15.

# 4 A NEW DATASET FOR TEMPORAL AND NON-TEMPORAL KNOWLEDGE BASE COMPLETION

A dataset based on Wikidata was proposed by García-Durán et al. (2018). However, upon inspection, this dataset contains numerical data as entities, such as ELO rankings of chess players, which are not representative of practically useful link prediction problems. Also, in this dataset, temporal informations is specified in the form of "OccursSince" and "OccursUntil" statements appended to triples, which becomes unwieldy when a predicate holds for several intervals in time. Moreover, this dataset contains only $11k$ entities and $150k$ which is insufficient to benchmark methods at scale.

The GDelt dataset described in Ma et al. (2018); Goel et al. (2020) holds many triples ($2M$), but does not describe enough entities ($500$). In order to adress these limitations, we created our own dataset from Wikidata, which we make available along with the code for this paper at `https://github.com/facebookresearch/tkbc`.

Starting from Wikidata, we removed all entities that were instance of scholarly articles, proteins and others. We also removed disambiguation, template, category and project pages from wikipedia. Then, we removed all facts for which the object was not an entity. We iteratively filtered out entities that had degree at least $5$ and predicates that had at least $50$ occurrences. With this method, we obtained a dataset of $432715$ entities, $407$ predicates and $1724$ timestamps (we only kept the years). Each datum is a triple (subject, predicate, object) together a timestamp range (begin, end) where begin, end or both can be unspecified. Our train set contains $7M$ such tuples, with about $10\%$ partially specified temporal tuples. We kept a validation and test set of size $50k$ each.

At train and test time, for a given datum (subject, predicate, object, [begin, end]), we sample a timestamp (appearing in the dataset) uniformly at random, in the range [begin, end]. For datum without a temporal range, we sample over the maximum date range. Then, we rank the objects for the partial query (subject, predicate, ?, timestamp).

# 5 EXPERIMENTAL RESULTS

## 5.1 EXPERIMENTAL SET-UP

We follow the experimental set-up in García-Durán et al. (2018); Goel et al. (2020). We use models from García-Durán et al. (2018) and Goel et al. (2020) as baselines since they are the best performing algorithms on the datasets considered. We report the filtered Mean Reciprocal Rank (MRR) defined in Nickel et al. (2016b). In order to obtaiqn comparable results, we use Table 1 and dataset statistics to compute the rank for each (model, dataset) pair that matches the number of parameters used in Goel et al. (2020). We also report results at ranks 10 times higher. This higher rank set-up gives an estimation of the best possible performance attainable on these datasets, even though the dimension used might be impractical for applied systems. All our models are optimized with Adagrad (Duchi et al., 2011), with a learning rate of $0.1$, a batch-size of $1000$. More details on the grid-search, actual ranks used and hyper-parameters are given in Appendix 8.7.

|              | ICEWS14      | ICEWS15-05   | Yago15k        |
|--------------|--------------|--------------|----------------|
| TA           | 0.48         | 0.47         | 0.32           |
| DE-SimplE    | 0.53         | 0.51         | -              |
| ComplEx      | 0.47 (0.47)  | 0.49 (0.49)  | **0.35** (0.36) |
| TComplEx     | **0.56** (0.61) | 0.58 (0.66) | **0.35** (0.36) |
| TNTComplEx   | **0.56 (0.62)** | **0.60 (0.67)** | **0.35 (0.37)** |

Table 2: Results for TA (García-Durán et al., 2018) and DE-SimplE (Goel et al., 2020) are the best numbers reported in the respective papers. Our models have as many parameters as DE-SimplE. Numbers in parentheses are for ranks multiplied by 10.

| Reg.             | MRR  |
|------------------|------|
| No regularizer   | 0.62 |
| $\Delta^2$       | 0.63 |
| $\Delta^3$       | 0.63 |
| $\Delta^4$       | 0.64 |
| $\Omega^3$       | 0.65 |
| $\Omega^3 + \Lambda_4$ | **0.67** |

Table 3: Impact of regularizers on ICEWS05-15 for TNTComplEx.

We give results on 3 datasets previously used in the litterature : ICEWS14, ICEWS15-05 and Yago15k. The ICEWS datasets are samplings from the Integrated Conflict Early Warning System (ICEWS)(Boschee et al., 2015)[1].García-Durán et al. (2018) introduced two subsampling of this data, ICEWS14 which contains all events occuring in 2014 and ICEWS05-15 which contains events occuring between 2005 and 2015. These datasets immediately fit in our framework, since the timestamps are already discretized.

The Yago15K dataset (García-Durán et al., 2018) is a modification of FB15k (Bordes et al., 2013) which adds "occursSince" and "occursUntil" timestamps to each triples. Following the evaluation setting of García-Durán et al. (2018), during evaluation, the incomplete triples to complete are of the form (subject, predicate, ?, occursSince | occursUntil, timestamp) (with reciprocal predicates). Rather than deal with tensors of order 5, we choose to unfold the (occursSince, occursUntil) and the predicate mode together, multiplying its size by two.

Some relations in Wikidata are highly unbalanced (eg. (?, InstanceOf, Human)). For such relations, a ranking evaluation would not make much sense. Instead, we only compute the Mean Reciprocal Rank for missing right hand sides, since the data is such that highly unbalanced relations happen on the left-hand side. However, we follow the same training scheme as for all the other dataset, including reciprocal relations in the training set. The cross-entropy loss evaluated on $400k$ entities puts a restriction on the dimensionality of embeddings at about $d = 100$ for a batch-size of 1000. We leave sampling of this loss, which would allow for higher dimensions to future work.

## 5.2 RESULTS

We compare ComplEx with the temporal versions described in this paper. We report results in Table 2. Note that ComplEx has performances that are stable through a tenfold increase of its number of parameters, a rank of 100 is enough to capture the static information of these datasets. For temporal models however, the performance increases a lot with the number of parameters. It is always beneficial to allow a separate modeling of non-temporal predicates, as the performances of TNTComplex show. Finally, our model match or beat the state of the art on all datasets, even at identical number of parameters. Since these datasets are small, we also report results for higher ranks (10 times the number of parameters used for DE-SimplE).

On Wikidata, $90\%$ of the triples have no temporal data attached. This leads to ComplEx outperforming all temporal models in term of average MRR, since the Non-Temporal MRR (NT-MRR) far

---

[1]More information can be found at http://www.icews.com

|            | MRR  | NT-MRR | T-MRR |
|------------|------|--------|-------|
| ComplEx    | **0.45** | **0.48** | 0.29  |
| TComplEx   | 0.42 | 0.45   | 0.30  |
| TNTComplEx | 0.44 | 0.47   | **0.32** |

Table 4: Results on wikidata for entity dimension $d = 100$.

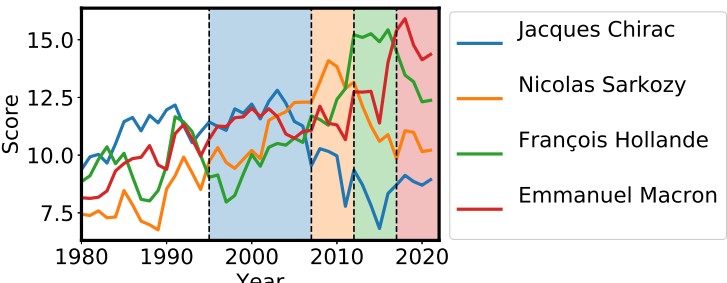

Figure 2: Scores for triples (President of the French republic, office holder, {Jacques Chirac | Nicolas Sarkozy | François Hollande | Emmanuel Macron}, $[1980, 2020]$)

outweighs the Temporal MRR (T-MRR). A breakdown of the performances is available in table 4. TNTComplEx obtains performances that are comparable to ComplEx on non-temporal triples, but are better on temporal triples. Moreover, TNTComplEx can minimize the temporal cross-entropy (7) and is thus more flexible on the queries it can answer.

Training TNTComplEx on Wikidata with a rank of $d = 100$ with the full cross-entropy on a Quadro GP 100, we obtain a speed of $5.6k$ triples per second, leading to experiments time of $7.2$ hours. This is to be compared with $5.8k$ triples per second when training ComplEx for experiments time of $6.9$ hours. The additional complexity of our model does not lead to any real impact on runtime, which is dominated by the computation of the cross-entropy over $400k$ entities.

## 6  QUALITATIVE STUDY

The instantaneous loss described in equation (1), along with the timestamp sampling scheme described in the previous section only enforces correct rankings along the "object" tubes of our order-$4$ tensor. In order to enforce a stronger temporal consistency, and be able to answer queries of the type (subject, predicate, object, ?), we propose another cross-entropy loss along the temporal tubes:

$$\tilde{\ell}(\hat{X}; (i, j, k, l)) = -\hat{X}_{i,j,k,l} + \log \Big( \sum_{l'} \exp \big( \hat{X}_{i,j,k,l'} \big) \Big). \tag{7}$$

We optimize the sum of $\ell$ defined in Equation 1 and $\tilde{\ell}$ defined in Equation 7. Doing so, we only lose 1 MRR point overall. However, we make our model better at answering queries along the time axis. The macro area under the precision recall curve is $0.92$ for a TNTComplEx model learned with $\ell$ alone and $0.98$ for a TNTComplEx model trained with $\ell + \tilde{\ell}$.

We plot in Figure 2 the scores along time for train triples (president of the french republic, office holder, {Jacques Chirac | Nicolas Sarkozy | François Hollande | Emmanuel Macron}, $[1980, 2020]$). The periods where a score is highest matches closely the ground truth of start and end dates of these presidents mandates which is represented as a colored background. This shows that our models are able to learn rankings that are correct along time intervals despite our training method only ever sampling timestamps within these intervals.

## 7  CONCLUSION

Tensor methods have been successful for Knowledge Base completion. In this work, we suggest an extension of these methods to Temporal Knowledge Bases. Our methodology adapts well to the various form of these datasets : point-in-time, beginning and endings or intervals. We show that our methods reach higher performances than the state of the art for similar number of parameters. For several datasets, we also provide performances for higher dimensions. We hope that the gap between low-dimensional and high-dimensional models can motivate further research in models that have increased expressivity at lower number of parameters per entity. Finally, we propose a large scale temporal dataset which we believe represents the challenges of large scale temporal completion in knowledge bases. We give performances of our methods for low-ranks on this dataset. We believe that, given its scale, this dataset could also be an interesting addition to non-temporal knowledge base completion.

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

# 8 APPENDIX

## 8.1 UNFOLDING AND THE CP DECOMPOSITION

Let $X = [\![U, V, W, T]\!]$, that is $X_{i,j,k,l} = \langle u_i, v_j, w_k, t_l \rangle$. Then according to Kolda & Bader (2009), unfolding along modes 3 and 4 leads to an order three tensor of decomposition $\tilde{X} = [\![U, V, W \circ T]\!]$. Where $\circ$ is the Khatri-Rao product (Smilde et al., 2005), which is the column-wise Kronecker product : $W \circ T = (W_{:,1} \otimes T_{:,1}, \ldots, W_{:,R} \otimes T_{:,R})$.

Note that for a fourth mode of size $L$: $(W \circ T)_{L(k-1)+l} = w_k \odot t_l$. This justifies the regularizers used in Section 3.2.

## 8.2 TEMPORAL REGULARIZER AND NUCLEAR NORMS

Consider the penalty:

$$\Omega(U, V, W, T) = \frac{1}{4} \left( \|U\|_4^4 + \|V\|_4^4 + \|W\|_4^4 + \|T\|_4^4 + \alpha\|T_{1:} - T_{:-1}\|_4^4 \right)$$

Let us define a new norm on vectors:

$$\|t\|_{\tau 4} = \left( \|t\|_4^4 + \alpha\|t_{1:} - t_{:-1}\|_4^4 \right)^{1/4}$$

$\| \cdot \|_{\tau 4}$ is a norm and lets us rewrite:

$$\Omega(U, V, W, T) = \sum_{r=1}^{R} \frac{1}{4} \left( \|u_r\|_4^4 + \|v_r\|_4^4 + \|w_r\|_4^4 + \|t_r\|_{\tau 4}^4 \right).$$

Following the proof in Lacroix et al. (2018) which only uses homogeneity of the norms, we can show that $\Omega(U, V, W, T)$ is a variational form of an atomic norm with atoms :

$$\mathcal{A} = \{u \otimes v \otimes w \otimes t \mid \|u\|_4, \|v\|_4, \|w\|_4 \leq 1 \text{ and } \|t\|_{\tau 4} \leq 1\}$$

## 8.3 NUCLEAR NORMS ON UNFOLDINGS

We consider the regularizer :

$$\Omega^{N3}(U, V, T; (i, j, k, l)) = \frac{1}{3} \left( \|u_i\|_3^3 + \|u_k\|_3^3 + \|v_k \odot t_l\|_3^3 \right).$$

Let $D^{\text{subj}}$ (resp. obj, pred/time) the diagonal matrix containing the cubic-roots of the marginal probabilities of each subject (resp. obj, pred/time) in the dataset. We denote by $\circ$ the Kathri-Rao product between two matrices (the columnwise Kronecker product). Summing over the entire dataset, we obtain the penalty:

$$\frac{1}{|S|} \sum_{(i,j,k,l) \in S} \Omega^{N3}(U, V, T; (i, j, k, l)) = \frac{1}{3} \left( \|D^{\text{subj}}U\|_3^3 + \|D^{\text{obj}}U\|_3^3 + \|D^{\text{pred/time}}(V \circ T)\|_3^3 \right).$$

Dropping the weightings to simplify notations, we state the equivalence between this regularizer and a variational form of the nuclear 3-norm of an order 4 tensor:

$$\inf_{[U_1, U_2, U_3, U_4] = X} \frac{1}{3} \left( \sum_{r=1}^{R} \|u_r^{(1)}\|_3^3 + \|u_r^{(2)}\|_3^3 + \|u_r^{(3)} \otimes u_r^{(4)}\|_3^3 \right) = \inf_{[U_1, U_2, U_3, U_4] = X} \frac{1}{3} \left( \sum_{r=1}^{R} \prod_{d=1}^{4} \|u_r^{(d)}\|_3 \right).$$

The proof follows Lacroix et al. (2018), noting that $\|u_r^{(3)} \otimes u_r^{(4)}\|_3^3 = \|u_r^{(3)}\|_3^3 \|u_r^{(4)}\|_3^3$. Note that for $D^{\text{pred/time}} = D^{\text{pred}} D^{\text{time}}$, there would also be equality of the weighted norms. However, in the application considered, time and predicate are most likely not independent, leading to different weightings of the norms.

## 8.4 DATASET STATISTICS

Statistics of all the datasets used in this work are gathered in Table 5.

|            | ICEWS14 | ICEWS05-15 | Yago15k | Wikidata |
|------------|---------|------------|---------|----------|
| Entities   | 6869    | 10094      | 15403   | 432715   |
| Predicates | 460     | 502        | 102     | 814      |
| Timestamps | 365     | 4017       | 170     | 1726     |
| |S|        | 72826   | 368962     | 110441  | 7224361  |

Table 5: Dataset statistics

## 8.5 DETAILED RESULTS

|                | ICEWS14 | | | | ICEWS15-05 | | | | Yago15k | | | |
|----------------|------|------|------|------|------|------|------|------|------|------|------|------|
|                | MRR | H@1 | H@3 | H@10 | MRR | H@1 | H@3 | H@10 | MRR | H@1 | H@3 | H@10 |
| TA             | 0.48 | 0.37 | -    | 0.69 | 0.47 | 0.35 | -    | 0.73 | 0.32 | 0.23 | -    | 0.51 |
| DE-SimplE      | 0.53 | 0.42 | 0.59 | 0.73 | 0.51 | 0.39 | 0.58 | 0.75 | -    | -    | -    | -    |
| ComplEx        | 0.47 | 0.35 | 0.53 | 0.70 | 0.49 | 0.37 | 0.55 | 0.72 | **0.35** | **0.28** | **0.35** | **0.52** |
| TComplEx       | **0.56** | **0.47** | **0.61** | 0.73 | 0.58 | 0.49 | 0.64 | 0.76 | **0.35** | 0.27 | **0.36** | **0.52** |
| TNTComplEx     | **0.56** | 0.46 | **0.61** | **0.74** | **0.60** | **0.50** | **0.65** | **0.78** | **0.35** | **0.28** | 0.35 | **0.52** |
| ComplEx (x10)  | 0.47 | 0.35 | 0.54 | 0.71 | 0.49 | 0.37 | 0.55 | 0.73 | 0.36 | **0.29** | 0.36 | **0.54** |
| TComplEx (x10) | 0.61 | **0.53** | **0.66** | **0.77** | 0.66 | **0.59** | **0.71** | 0.80 | 0.36 | 0.28 | 0.38 | **0.54** |
| TNTComplEx (x10) | **0.62** | 0.52 | **0.66** | 0.76 | **0.67** | **0.59** | **0.71** | **0.81** | **0.37** | **0.29** | **0.39** | **0.54** |

Table 6: Results for TA (García-Durán et al., 2018) and DE-SimplE (Goel et al., 2020) are the best numbers reported in the respective papers.

## 8.6 STANDARD DEVIATIONS

We give the standard deviations for the MRR computed over 5 runs of TNTComplEx on all datasets:

|            | ICEWS14 | ICEWS15-05 | Yago15k | Wikidata (T) | Wikidata (NT) |
|------------|---------|------------|---------|--------------|---------------|
| TNTComplEx | 0.0016  | 0.0011     | 0.00076 | 0.0035       | 0.0012        |

## 8.7 GRID SEARCH

For ICEWS14, ICEWS05-15 and Yago15k, we follow the grid-search below :

Using Table 1 to compute the number of parameters and the dataset statistics in Table 5, we use the following ranks to match the number of parameters of DE-SimplE in dimension 100:

|            | ICEWS14 | ICEWS05-15 | Yago15k |
|------------|---------|------------|---------|
| DE-SimplE  | 100     | 100        | 100     |
| ComplEx    | 182     | 186        | 196     |
| TComplEx   | 174     | 136        | 194     |
| TTComplEx  | 156     | 128        | 189     |

