# OpenReview forum: "Tensor Decompositions for Temporal Knowledge Base Completion"
_ICLR.cc/2020/Conference — Accept (Poster)_

### Official Review · AnonReviewer1 · 2019-10-16
**Official Blind Review #1**

**Rating:** 6

**Review:**

In this paper, the authors study an important problem, i.e., time-aware link prediction in a knowledge base. Specifically, the authors focus on predicting the missing link in a quadruple, i.e., (subject, predicate, ?, timestamp). In particular, the authors design a new tensor (order 4) factorization based method with proper regularization terms shown in Eqs.(4-6).

The authors also prepare a dataset which may be helpful for further study on this topic, which is highly appreciated.

Empirical studies on three public datasets show the effectiveness of the proposed model over the state-of-the-art tensor factorization based methods that are published very recently.


Some comments:

1 The technical novelty is a bit limited in compared with those 3-order tensor factorization based methods.

2 For the time-aware link prediction problem, some deep learning based methods may also perform well, especially those capturing sequential signals, e.g., RNN and their variants. The authors are encouraged to include those works in the related work and in the empirical studies.

Minor:

Typo: 'ICEWS05-15'


**Experience Assessment:**

I have read many papers in this area.

**Review Assessment: Checking Correctness Of Derivations And Theory:**

I assessed the sensibility of the derivations and theory.

**Review Assessment: Checking Correctness Of Experiments:**

I assessed the sensibility of the experiments.

**Review Assessment: Thoroughness In Paper Reading:**

I read the paper at least twice and used my best judgement in assessing the paper.

---

> ### Author Response · Authors · 2019-11-09
> **detailed comments**
>
> 1 The technical novelty is a bit limited in compared with those 3-order tensor factorization based methods.
> → Despite a simple modification in the model, we believe our study of regularizers and their effects is non-trivial. Our model provide a large improvement over the previous state of the art for temporal knowledge base completion on all benchmarks considered.
>
> 2 For the time-aware link prediction problem, some deep learning based methods may also perform well, especially those capturing sequential signals, e.g., RNN and their variants. The authors are encouraged to include those works in the related work and in the empirical studies.
> →The work we include from Garcia-Duran et al. 2018 uses RNN. Its performances are included in Table 2 as TA. The other works we were able to find that use RNN are Lei et al. 2019 and Chen et al. 2018. They focus on much smaller graphs than ours: maximum number of node 256 for Lei et al., 274 for Chen et al., while we work with up to 400k entities.
>
> Lei et al. 2019 GCN-GAN: A Non-linear Temporal Link Prediction Model for Weighted Dynamic Networks
> Chen et al. 2018 GC-LSTM: Graph Convolution Embedded LSTM for Dynamic Link Prediction

---

### Official Review · AnonReviewer3 · 2019-10-23
**Official Blind Review #3**

**Rating:** 6

**Review:**

The work in this paper is focused on the task of knowledge base completion, dealing specifically with temporal relations, which is quite important in practice, and also not as well studied in literature. Specifically, the authors have 3 main contributions in this paper:
1. They present an order 4 tensor factorization for dealing with temporal data, which is a nice extension of the work in Lacroix 2018. The authors introduce different forms of regularization to extend to the order 4 tensors. Inspired by previous work, they produce different regularization strategies for order 4 tensor by unfolding the modes to reduce it ot an order 3 formulation. They also describe a regularization term for smoothing the temporal embeddings.
3. Finally, the authors mine wikidata for temporal relations and contribute a dataset based on wikidata that is much larger than existing datasets for this task.


Overall, I think this paper is  interesting as it provides an incremental extension of ComplEx to the temporal case, and the experiments to support the formulation show improvements in MRR. However, the experiments around standard benchmarks as well as the data produced by the authors do not always support the hypothesis that modeling temporal dimension in the proposed formulation is a win for the KB completion task.  For example, the use of auxiliary losses  for enforcing temporal constraints makes the overall performance worse. This is mentioned in a section, but I think it deserves a more thorough explanation. Also, there is no mention about statistical significance of the results, and so it is hard to judge the claim of these being SOTA as made by the authors. For example, on the wikidata dataset produced, the ComplEx model outperforms the proposed models.



Strengths:
1. The work in this paper is quite well motivated and the modeling formulation is clear and easy to follow. The authors did a great job in citing relevant work and walking through the model formulation as well as the various regularization terms introduced.
2. The authors compare their work to the previous SOTA - the ComplEx models for multiple datasets, including their own released datasets. They present a clear set of experiments to demonstrate the effectiveness of their approach both on non-temporal and temporal relations.
3. There is a dataset being released and also code, which should aid in reproducibility of the results (though I have not tested the code).

Areas to be addressed:
1. The work seems to assume that the time is discretized by year. However, it is unclear how one would deal with a KB where the temporal relations can change on the order of weeks (example popular movies). For that matter, how would the model be modified to deal with heterogenous time scales in the evolution of the relations in the KB? Can the authors add some clarification as well as explanations for how this would be addressed in this formulation?
2. While there is a section devoted to different regularization, from Table 3, the impact of choosing the right regularizer seems minimal in terms of MRR. Also, it seems that the results in Table 3 are comparable to the “ranks multiplied by 10” setting in Table 2. What is the reason for this choice?
3. For Table 4, why is there a performance difference between static and non-static relations? It would help if the authors could provide some more error analyses to dive into this performance difference - is it just data imbalance or inherent difficulty in the task for temporal relations.
4. Section 6 talks about enforcing another auxiliary loss, however, these results are not part of the table. I would urge the authors to add this to Table 4. Also, the loss in MRR mentioned is it the average loss or does enforcing this auxiliary loss also influence performance on T-MRR as well? If so, might it be that the auxiliary loss is too strong and might need to be penalized? Did the authors try penalizing the auxiliary loss? Finally, the graph in Figure 2 is hard to follow when printed in black and white. What would the plot look like in comparison to a model that does not enforce the auxiliary function for the same example? I would recommend re-working Section 6 to provide some more details about the performance of the models both across the various datasets as well as error analyses of a few examples compared across TNTComplex, TNTComplex + auxiliary loss + Baseline Complex. Having some representative examples would make it easy to understand where these models differ in their performance.
5. In the results, can the authors include metrics like filtered MRR as well as Hits@10, similar to the one suggested by Bordes 2013? This would make it easier to compare against previous literature results which all seem to be reporting on these metrics.


**Experience Assessment:**

I have read many papers in this area.

**Review Assessment: Checking Correctness Of Derivations And Theory:**

I assessed the sensibility of the derivations and theory.

**Review Assessment: Checking Correctness Of Experiments:**

I carefully checked the experiments.

**Review Assessment: Thoroughness In Paper Reading:**

I read the paper thoroughly.

---

> ### Author Response · Authors · 2019-11-09
> **Detailed comments**
>
> 1. The work seems to assume that the time is discretized by year. However, it is unclear how one would deal with a KB where the temporal relations can change on the order of weeks (example popular movies). For that matter, how would the model be modified to deal with heterogenous time scales in the evolution of the relations in the KB? Can the authors add some clarification as well as explanations for how this would be addressed in this formulation?
> → Changing the discretization of time is not an issue unless the number of timestamps to be considered becomes too large. For the exemple of movies, with 52 weeks per year, this would lead to 5200 timestamps for 100 years worth of movies, which is completely manageable. As mentioned in our answer to Reviewer #2 however, the temporal regularizer becomes more important when the granularity along time increases.
> Regarding heterogeneous timescales : in the temporal regularizer, we penalize the difference of embeddings of successive timestamps, irrespective of the actual timestamps. In preliminary experiments, we also tried weighting by the time difference: ||T_i - T_{i+1}||_2^2 / (ts_{i+1} - ts_i), where T_i is the embedding corresponding to timestamp ts_i. This reduces the temporal regularization for embeddings of timestamps that are far from their neighbors. This did not lead to improvements on the datasets considered.
>
> 2. While there is a section devoted to different regularization, from Table 3, the impact of choosing the right regularizer seems minimal in terms of MRR. Also, it seems that the results in Table 3 are comparable to the “ranks multiplied by 10” setting in Table 2. What is the reason for this choice?
> → Impact of regularizer: the impact is e.g., 2 points of MRR absolute on ICEWS-5, which is substantial.  We consider our results with higher ranks as the new state-of-the-art (since performances are significantly better than previous methods), so we compare the regularizers in Table 3 in this most challenging and important setup.
>
> 3. For Table 4, why is there a performance difference between static and non-static relations? It would help if the authors could provide some more error analyses to dive into this performance difference - is it just data imbalance or inherent difficulty in the task for temporal relations.
> → The difference in performance is due to a discrepancy in temporal vs non-temporal data. Notably, 93% of temporal tuples (subject, predicate) have two or more valid objects. This proportion is only 47% for non-temporal tuples.  One-to-many relations lead to lower MRR than one-to-one relations (see table 3, Lacroix et al. 2018 Canonical Tensor Decomposition for Knowledge Base Completion)
>
> 4. Section 6 talks about enforcing another auxiliary loss, however, these results are not part of the table. I would urge the authors to add this to Table 4. Also, the loss in MRR mentioned is it the average loss or does enforcing this auxiliary loss also influence performance on T-MRR as well? If so, might it be that the auxiliary loss is too strong and might need to be penalized? Did the authors try penalizing the auxiliary loss?
>
> → The auxiliary loss has no influence on the T-MRR,  the point of MRR is lost over non-temporal triples. We did not try reducing the weight of this loss in the overall loss to obtain the best (MRR, time AUC) operating point as this section is meant to be an example of new metrics that could be interesting on this dataset.
>
> Finally, the graph in Figure 2 is hard to follow when printed in black and white. What would the plot look like in comparison to a model that does not enforce the auxiliary function for the same example?
>
> → We will add different strokes for each presidents in the final version of the paper.  On this example, the plot for TNTComplEx trained without the auxiliary loss gives similar scores to the presidents over time.
>
> Having some representative examples would make it easy to understand where these models differ in their performance.
> → The breakdown we provide shows that the difference comes from non-temporal triples, which is expected when comparing a temporal and non-temporal model. Since the non-temporal triples account for 90% of the link prediction loss, it is natural that adding a new loss will impact in majority the MRR of these triples and leave the MRR of temporal triples intact, which is what we observe.
>
> 5. In the results, can the authors include metrics like filtered MRR as well as Hits@10, similar to the one suggested by Bordes 2013? This would make it easier to compare against previous literature results which all seem to be reporting on these metrics.
> → We report the filtered MRR similarly to previous state of the art on these datasets. This is specified in the experimental set-up section. To avoid clutter, we added hits@k in the supplementary materials (Appendix 8.5).

---

### Official Review · AnonReviewer2 · 2019-10-24
**Official Blind Review #2**

**Rating:** 6

**Review:**

Review:
This paper extends the ComplEx model (Trouillon et al., 2016) for completing temporal knowledge bases by augmenting it with timestamp embeddings. Besides, based on the assumption that these timestamp representations evolve slowly over time, the paper introduces this prior as a regularizer. Also, the paper adds a non-temporal component to the model to deal with static facts in knowledge bases. The proposed model has been evaluated using the current benchmark temporal event datasets, showing state-of-the-art performance.
This paper could be weakly accepted because the paper introduces new regularization schemes based on the tensor nuclear norm for tensor decomposition over temporal knowledge graphs, which could be a significant algorithmic contribution. Additionally, the submission empirically studies the impact of regularizations for the proposed model, and weight different regularizers according to the joint marginal of timestamps and predicates, which achieves good experimental results.
Feedback to improve the paper:
1. For novelty, the paper does not clearly point out that Ma et al. (2018) already augmented the ComplEx model with time embedding for completing temporal knowledge graphs.
2. The paper does not point out whether the units of the timestamps affects the model and how to adjust the model accordingly. For example, the time granularity of the ICEWS dataset is 24 hours. If we switch the unit of timestamps from hours to days, do the results of the proposed model change? If yes, how to peak the best time unit for a given dataset?
3. For the Wikidata, the author reports the filtered Mean Reciprocal Rank of the conducted experiments, where the author provides not only the overall score but also the temporal MRR. However, the paper does not provide information about error bars as well as the unfiltered version of the experiment results. Since the results of TComplEx and TNTComplEx are only slightly better than ComplexE, the reviewer doubts whether the proposed model can really beat the ComplEx model when considering the error bars. Also, NT-MRR and T-MRR are not clearly defined.
4. The submission proposes a new large-scale temporal event dataset, and states that this dataset is more realistic than the current benchmarks. However, the reviewer does not find any argument in the paper to support this statement.
References:
Théo Trouillon, Johannes Welbl, Sebastian Riedel, Éric Gaussier, and Guillaume Bouchard. Com- plex embeddings for simple link prediction. In International Conference on Machine Learning, pp. 2071–2080, 2016.
Yunpu Ma, Volker Tresp, and Erik A Daxberger. Embedding models for episodic knowledge graphs. Journal of Web Semantics, pp. 100490, 2018.

**Experience Assessment:**

I have published one or two papers in this area.

**Review Assessment: Checking Correctness Of Derivations And Theory:**

I carefully checked the derivations and theory.

**Review Assessment: Checking Correctness Of Experiments:**

I carefully checked the experiments.

**Review Assessment: Thoroughness In Paper Reading:**

I read the paper at least twice and used my best judgement in assessing the paper.

---

> ### Author Response · Authors · 2019-11-09
> **detailed comments**
>
> 1. For novelty, the paper does not clearly point out that Ma et al. (2018) already augmented the ComplEx model with time embedding for completing temporal knowledge graphs.
> → We already cite Ma et al. in the related work section. Please note that they *do not* extend the Complex model. To the best of our understanding, this paper proposes an extension to Tucker in which a new core tensor is learned for every timestamps. This leads to many more parameters than this work. Their results are below the other results we include in our state-of-the-art comparison.
>
> 2. The paper does not point out whether the units of the timestamps affects the model and how to adjust the model accordingly. For example, the time granularity of the ICEWS dataset is 24 hours. If we switch the unit of timestamps from hours to days, do the results of the proposed model change? If yes, how to peak the best time unit for a given dataset?
> → At higher granularities, we expect the temporal regularizer to become even more useful, since it should mitigate the lack of data per timestamps. In practice, several granularities could be cross-validated. For ICEWS, we use the granularity of the evaluation proposed in previous work.
>
> 3. For the Wikidata, the author reports the filtered Mean Reciprocal Rank of the conducted experiments, where the author provides not only the overall score but also the temporal MRR. However, the paper does not provide information about error bars as well as the unfiltered version of the experiment results. Since the results of TComplEx and TNTComplEx are only slightly better than ComplexE, the reviewer doubts whether the proposed model can really beat the ComplEx model when considering the error bars. Also, NT-MRR and T-MRR are not clearly defined.
>
> → NT (NonTemporal-MRR) is the MRR computed over test triples that have no timestamps associated. T-MRR is the MRR computed over test triples that have at least 1 timestamp associated. We do not provide unfiltered metrics because these tend to saturate more quickly than filtered ones. The standard deviation for non-temporal queries is 0.0035 for TNTComplEx on non-temporal relations, making the 0.02 MRR difference we observe significant.
>
> 4. The submission proposes a new large-scale temporal event dataset, and states that this dataset is more realistic than the current benchmarks. However, the reviewer does not find any argument in the paper to support this statement.
> → We rephrased the conclusion of the paper. A knowledge base such as wikidata, freebase or yago all have timestamps naturally attached to the triples they represent. We believe this temporal data to be an essential part of the knowledge base and thus it should be included in knowledge base completion benchmarks. Moreover, we believe that the scale of this dataset is a good step in the direction of completion at the scale of full knowledge bases such as wikidata.

---

### Author Response · Authors · 2019-11-09
**General comments**

We thank the reviewers for their comments. We will address general concerns here and go into more details in separate comments. We added the standard deviation of TNTComplEx on each datasets, computed over 5 runs, to the supplementary materials (Appendix 8.5). The standard deviation is of the order of 1e-3, which is consistent with our use of boldface in results tables.
Regarding the performances of TNTComplEx compared to ComplEx on Wikidata:  we show that adding a new loss, our temporal model can answer queries that a non-temporal model cannot such as "When was Jacques Chirac the president of France ?". We further show that our TNTComplEx model is capable of answering these queries, while maintaining an overall MRR that is competitive with ComplEx.

---

### Decision · Program_Chairs · 2019-12-19

**Decision:**

Accept (Poster)

**Comment:**

The authors propose a new algorithm based on tensor decompositions for the problem of knowledge base completion. They also introduce new regularisers to augment their method. They also propose an new dataset for temporal KB completion.

All the reviews agreed that the paper addresses an important problem and presents interesting results. The authors diligently responded to reviewer queries and addressed most of the concerns raised by the reviewers.

Since all the reviewers are in agreement, I recommend that this paper be accepted.